# Prospective cohort study of surgical site infections following single dose antibiotic prophylaxis in caesarean section at a tertiary care teaching hospital in Medchal, India

Kalpana Basany[1]*, Sirshendu Chaudhuri[2], Lakshmi Shailaja P.[3], Varun Agiwal[2], Neelima Angaali[4], Nirupama A. Y.[2], Shailendra D.[5], Catherine Haggerty[6], P. S. Reddy[1,7]

1 Department of Obstetrics and Gynecology, Society for Health Allied Research and Education, INDIA MediCiti Institute of Medical Sciences, Hyderabad, Telangana, India, 2 Department of Epidemiology, Indian Institute of Public Health, Hyderabad, Telangana, India, 3 Department of Obstetrics and Gynecology, Fernandez Hospital, Hyderabad, Telangana, India, 4 Department of Microbiology, Nizams Institute of Medical Sciences, Hyderabad, Telangana, India, 5 Department of Pharmacology, SHARE INDIA, MediCiti Institute of Medical Sciences, Hyderabad, Telangana, India, 6 Department of Epidemiology, University of Pittsburgh, Pittsburgh, PA, United States of America, 7 Division of Cardiology, Department of Medicine, University of Pittsburgh School of Medicine, University of Pittsburgh, Pittsburgh, PA, United States of America

* kalpanabasany@gmail.com

## Abstract

### Background

Caesarean section (CS) is considered to be a life-saving operative intervention for women and new-borns in certain antepartum and intrapartum conditions. Caesarean delivery may be accompanied by several complications including surgical site infections (SSI). However, there is a significant lack of uniformity in the administration of antibiotics for preventing surgical site infections (SSI) following caesarean deliveries. The present study was conducted to determine the incidence of post CS SSI following the adoption of single-dose antibiotic prophylaxis as recommended by WHO at a tertiary care teaching hospital in Medchal, India. Also, to identify the risk factors of SSI and reported the bacteriological profiles and the antimicrobial susceptibility pattern of the culture positive isolates.

### Main objectives

To estimate the incidence of surgical site infections (SSI's) according to CDC criteria following WHO-recommended single-dose antibiotic prophylaxis for caesarean section at a tertiary care teaching hospital in Medchal, India.

### Methods

A prospective hospital-based study was conducted between June 2017 and December 2019, in which women who underwent caesarean delivery were followed up for 30 days post-delivery. Clinical details were collected using a structured questionnaire, and participants were followed up weekly after discharge to document any signs and symptoms of SSI. Symptomatic patients were requested to come to the hospital for further investigation

**Data Availability Statement:** All relevant data are within the manuscript and its Supporting Information files.

**Funding:** The author(s) received no specific funding for this work.

**Competing interests:** The authors have declared that no competing interests exist.

and treatment. Standard microbiological tests were conducted to detect microorganisms and their antibiotic sensitivity.

## Results

The study included 2,015 participants with a mean age of 24.1 years. The majority were multigravida (n = 1,274, 63.2%) and underwent emergency caesarean delivery (n = 1,232, 61.1%). Ninety two participants (4.6%, 95% CI: 3.7% to 5.6%) developed surgical site infections, with 91 (98.9%) having superficial and 1 (1.1%) having a deep infection. Among those who developed an SSI, 84 (91.3%) did so during their hospital stay, while 8 (8.7%) developed an SSI at home. The adjusted relative risk (a RR) for developing an SSI was 2.5 (95% CI: 1.4 to 4.6; power 99.9%) among obese women and 2.3 (95% CI: 1.1 to 4.7; power 100%) among women aged 25 years or younger. Microbial growth in culture was observed from 55 (75.8%) out of total 66 samples. The most common organisms identified were *Staphylococcus aureus* (n = 7(12.3%)23, 46.0%), *Klebsiella sp*. (n = 13, 26.0%), and *Escherichia coli* (n = 12, 24.0%).

## Conclusion

The rate of SSI following caesarean deliveries subjected to single dose antibiotic prophylaxis was low. Young women and obese women were at high risk of developing SSI.

## Introduction

Caesarean section (CS) is a considered to be a life-saving operative intervention for women and new-borns in certain antepartum conditions. Globally, CS is the most common major surgical intervention in pregnancies [1]. As per the World Health Organization (WHO), CS rate between 10–15% is considered optimum at the population level. Globally, the reported rate of CS is 21.1% and varies between 4.1% in West and Central Africa and 44.3% in the Latin America and the Caribbean region and increasing by 4% annually [2, 3]. In India, the overall CS rate has increased from 8.5% to 21.5% in the last 15 years [4]. According to the last national-level survey, a substantial variation exists between the states- from as low as 5.2% in Nagaland to as high as 60.7% in Telangana [4].

As a surgical procedure, caesarean delivery may be accompanied by several complications including surgical site infections (SSI) [5]. SSIs increase the morbidity and mortality of the mothers and babies and increases the length of hospital stay and thereby the cost of care [6, 7]. To prevent SSI, the WHO recommend using a single dose antibiotic, mostly the first generation cephalosporins or pencillin before 30 to minutes of incision for all women undergoing CS [8].

According to reports, the incidence of post-CS SSI worldwide is between 0.63% and 9.85%. [9–13] whereas in India, it ranges from 3.1% to 24.2% [7, 14–16]. However, there is a significant lack of uniformity in the administration of antibiotics for preventing surgical site infections (SSI) following caesarean deliveries.

Therefore, the present study was conducted to determine the incidence of post CS SSI following the adoption of single-dose antibiotic prophylaxis as recommended by WHO [8] in a tertiary care teaching hospital in Medchal, Telangana state, India. Prior to 2017, the hospital's practice involved administering post-operative antibiotics to all patients undergoing a

caesarean section (CS). However, after adapting the single-dose antibiotic prophylaxis, the administration of post-operative antibiotics was limited to cases where there was clinical or microbiological evidence of infection, specifically in cases of surgical site infections (SSI's). Additionally, we looked at the risk factors of SSI and reported the bacteriological profiles and the antimicrobial susceptibility pattern of the culture positive isolates.

## Methods

### Setting

The study was carried out at MediCiti Institute of Medical Sciences which is a tertiary care teaching hospital at Ghanpur village, located about 25km from the city of Hyderabad, in the Indian state of Telangana. The hospital has 570 inpatient beds out of which 60 were allocated to department of Obstetrics and Gynaecology. The caesarean section rate was 46% during the study period.

### Study design

Prospective study.

### Study duration

The study was conducted between June 2017 & December 2019.

### Study population

All women who underwent CS in the hospital during the study period were eligible for the study. We classified CS into two types—emergency and elective. And comparative analysis was made between emergency and elective caesarean section for various parameters (Table 1). An emergency CS is where there is either maternal or foetal compromise or where there is immediate threat to the life of women or foetus. Rest of the caesarean deliveries were grouped under elective CS. We excluded those women who delivered at other hospitals and came to the study hospital with surgical infection, those who died during CS or immediately after CS, and those who did not consent to participate.

### Sample size

For estimating 2% or less incidence of SSI after CS, we estimated our minimum required sample size to be 1,500 women who undergo CS; considering $\alpha = 0.05$, $\beta = 0.8$, & margin of error = 0.01.

### Antibiotic policy of the hospital

A single dose of prophylactic antibiotic was administered 30 min to one hour before the caesarean section. As a prophylaxis, we used injection cefazolin 1 gram. When cefazolin was unavailable, injection ampicillin 1 gram, or injection cefotaxime 1 gram was given. Post-operative antibiotics were not prescribed unless there was any clinical or bacteriological evidence of infection.

### Definition of variables

SSI was defined as "An infection of the superficial or deep skin incision, or of an organ or space, occurring up to 30 days after surgery" [17]. There are two types of SSI- superficial and deep. SSI was defined as per the CDC- National Healthcare Safety Network guidelines.

**Table 1. Profile of the study participants based on caesarean section.**

| Variables | Emergency CS, n = 1232 (%) | Elective CS, n = 783(%) | P-value |
|---|---|---|---|
| Age, years | | | |
| < = 25 | 908(73.7) | 507(64.8) | <0.001 |
| >25 | 324(26.3) | 280(35.5) | |
| BMI*, Kg/m$^2$ (n = 1,356) | | | |
| Underweight | 147(11.9) | 80(10.2) | 0.075 |
| Normal | 374(30.4) | 230(29.4) | |
| Overweight | 147(11.9) | 74(9.5) | |
| Obesity | 171(13.9) | 133(17) | |
| Current pregnancy registered | 1086(88.1) | 717(91.6) | 0.015 |
| Number of prenatal visits to hospital≥ 4 times | 968(78.6) | 645(82.4) | 0.411 |
| Presence of anaemia | 94(7.6) | 87(11.1) | 0.008 |
| Presence of hypertension | 207(16.8) | 94(12) | 0.003 |
| Presence of hypothyroidism | 171(13.9) | 97(12.4) | 0.337 |
| Presence of gestational diabetes | 48(3.9) | 33(4.2) | 0.723 |
| Previous caesarean section | 437(35.5) | 687(87.7) | <0.001 |
| Duration of labour (Hours) (n = 2,015) | | | |
| Non-labour | 118 (9.6) | 783 (100) | <0.001 |
| <6 | 597(48.4) | 0 | |
| 6–12 | 381(30.9) | 0 | |
| >12 | 136(11) | 0 | |

All the suspected infections were confirmed either clinically by the treating gynaecologist or microbiologically by the culture results.

Body mass index (BMI) was calculated by weight in Kg divided by height in meter$^2$. If height and weight measures were missed in the first trimester, or the participants recruited in the second trimester or later, BMI was excluded in the analysis. BMI was categorized according to underweight (BMI < 18.5 kg/m$^2$), normal (BMI 18.5 to 22.93 kg/m$^2$), overweight (BMI 23 to 24.9 kg/m$^2$, and obese (BMI ≥25 kg/m$^2$) [18]. After admission, all women were assessed for co-morbid medical conditions like diabetes, hypertension, and thyroid disorders.

### Data collection

All pertinent clinical details were collected from the participants hospital record using structured questionnaire by a dedicated study nurse. The questionnaire and the dataset were uploaded as supplementary files.

### Post discharge follow up

The hospital policy was to discharge the patient after suture removal on 6[th] or 7[th] postoperative day. The duration of hospital stay is from the day of admission to hospital to till the day of discharge from hospital. At the time of discharge from the hospital, the women were educated about the signs and symptoms of wound infection and were asked to report to the study nurse on noticing any signs and symptoms. The participants' contact number(s) were collected before discharge. Post-discharge, the information on signs and symptoms were also collected by weekly telephone calls till the end of 30[th] day from the date of operation by the study nurse. A standard script and a questionnaire in local language was used to enquire about the general health and wound infection. If suspected to have SSI, the women were followed up at the outpatient department (OPD).

### Laboratory diagnosis

In women where there was pus or discharge from the wound, two swabs were taken. Direct microscopy was done using the first swab, a smear was made on a clean glass slide and stained by Gram's stain. The second swab was inoculated onto plates of 5% sheep blood agar & Mac Conkey agar by rolling the swab over the agar and streaking from the primary inoculums, using a sterile bacteriological loop. If growth was seen after 24hrs, standard biochemical reactions were performed to isolate the organism. Antibiotic susceptibility testing (Antibiogram) of above isolates was performed by Kirby-Bauer disc diffusion method, using Muller Hinton agar plates according to Central Laboratory Standards Institute guidelines (CLSI). The same microbiological procedure was followed for those who were discharged and followed up in the OPD.

### Statistical analysis

The data was analyzed by STATA version 14.0. Descriptive statistics in terms of mean and standard deviation (SD) or median and interquartile range (IQR) were used to characterize the participants. Incidence of SSI was calculated in terms of percentage with 95% confidence interval (CI). Unpaired t-test was applied to test the difference in mean between two groups whereas association between two categorical variables was found out using chi-square test. Univariate logistic regression model was conducted to assess the determinants with a p-value lower than 0.2 were selected for inclusion in the final regression model, which was then analyzed using multiple logistic regression method. Risk of the predictors was estimated by adjusted relative risk (a RR) with 95% CI. A p-value <0.05 was considered significant for all statistical tests. Descriptive statistics were used to report the bacteriological profile of the SSI. Authors have access to deidentified data.

### Human subject protection

The study was approved by the MediCiti Ethics Committee of the institute which states that "on consideration of the study papers submitted, permission is also accorded for publication of any paper on the studying subjects." Besides, written informed consent was taken from all the participants.

## Results

We recruited a total number of 2,038 participants in the study. We excluded 23 participants due to missing information. Finally, 2,015 participants' information were available for analysis. The mean age of the participants was 24.1 years (SD 3.2 years), and majority were multigravida (n = 1,274, 63.2%) (Table 2). Seven hundred eleven (35.3%) participants had at least one known comorbidity Twelve hundred and thirty two (61.1%) participants underwent emergency CS. The average hospital stay was 9.4 days (SD 3.2 days).

### Incidence of SSI

Surgical site infection was developed in 92 (4.6%, 95% CI: 3.7 to 5.6%) out of 2015 participants. Out of this 91 (98.9%) had superficial and 1 (1.1%) had deep infection. Eighty-four (91.3%) participants developed infection during the hospital stay and eight (8.7%) at home. The infection rate was higher among women underwent emergency CS (n = 70, 5.7%; 95% CI: 4.5 to 7.1) compared to those who underwent elective CS (n = 22, 2.8%; 95% CI: 1.8 to 4.2).

**Table 2. Clinical profile of the participants, Medchal, Telangana, India (n = 2,015).**

| Variables | Frequency (%) |
|---|---|
| Age, years | |
| < = 25 | 1,415 (70.2) |
| >25 | 600 (29.8) |
| BMI*, Kg/m$^2$ (n = 1,356) | |
| Underweight | 227 (16.7) |
| Normal | 604 (44.5) |
| Overweight | 221 (16.3) |
| Obesity | 304 (22.4) |
| Current pregnancy registered | 1803 (89.5) |
| Number of prenatal visits to hospital≥ 4 times | 1613 (80.1) |
| Presence of anaemia (< 11gm%) | 181 (9.0) |
| Presence of hypertension BP ≥140 /90 mmHg) | 301 (14.9) |
| Presence of hypothyroidism | 268 (13.3) |
| Presence of gestational diabetes | 81 (4.0) |
| Previous caesarean section | 1,124 (55.8) |
| Duration of labour (Hours) (n = 2,015) | |
| Non-labour | 901 (44.7) |
| <6 | 597 (29.6) |
| 6–12 | 381 (18.9) |
| >12 | 136 (6.8) |

*First trimester BMI is unavailable for 659 participants

The median time to develop SSI was 7 days (IQR 6 to 7 days). The average hospital stay was 11.8 days (SD 3.7 days) in women who developed SSI and 9.3 (SD 3.1) days without SSI and mean difference was 2.5 days (95% CI: 1.9 to 3.2 days; p <0.001).

## Risk factors

Adjusted relative risk (a RR) of SSI was 2.3 times higher (95% CI: 1.1 to 4.8; Power 100%) among young age (< = 25 years) and 2.5 times higher (95% CI: 1.4 to 4.6; Power 99.9%) among obese women. Women who underwent emergency LSCS (a RR 3.0 95% CI: 1.1 to 8.8, p = 0.06) had a higher SSI rate, though statistically not significant (Table 3).

## Bacteriological profile

A total number of 66 wound discharge samples were sent for bacteriological assessment, out of which growth yielded in 50 (75.8%) samples, seven samples (14.0%) grew multiple organisms thereby yielding a total of 57 isolates. We found, *Klebsiella sp.* (n = 13, 22.8%), and *Escherichia coli* (n = 12, 21.1%) and *Staphylococcus aureus* (n = 7, 12.3%) being the most common organisms (Table 4).

Staphylococcus aureus was commonly sensitive to linezolid (6 isolates, 85.7%), and levofloxacin). All the isolates were sensitive to methicillin and vancomycin (Table 4). The isolates were commonly resistant to penicillin (six isolates, 85.7%) (Table 4). Among the gram-negative organisms, Klebsiella was the commonest organism. It was sensitive to ceftazidime and clavulanic acid combination (13 isolates, 100%), imipenem (10 isolates, 76.9%), and cefepime (9 isolates, 69.2%) and commonly resistant to ampicillin (11 isolates, 84.6%) and amoxycillin/clavulanic acid combination (10 isolates, 76.9%).

Table 3. Associated socio-demographic and clinical factors for developing the SSI.

| Variable | Reference group | Study group | Univariate Logistic Regression | | Multivariate Logistic Regression | |
|---|---|---|---|---|---|---|
| | | | RR (95% CI) | P-value | a RR (95% CI) | P-value |
| Age (years) | >25 | < = 25 | 1.8 (1.1–3.0) | 0.030 | 2.3 (1.1–4.8) * | 0.026 |
| BMI | Normal | Underweight | 0.7 (0.3–1.8) | 0.479 | 0.7 (0.3–1.7) | 0.413 |
| | | Overweight | 1.4 (0.7–2.9) | 0.388 | 1.3 (0.6–2.8) | 0.465 |
| | | Obesity | 2.4 (1.3–4.3) | 0.004 | 2.5 (1.4–4.6)* | 0.003 |
| Parity | Primi | Multi | 0.5 (0.3–0.7) | <0.001 | 0.6 (0.3–1.2) | 0.147 |
| Duration of labor (hours) | Non-labor | <6 | 1.5 (0.9–2.6) | 0.142 | 0.5 (0.2–1.2) | 0.127 |
| | | 6–12 | 2.1 (1.2–3.8) | 0.009 | 0.6 (0.2–1.7) | 0.328 |
| | | >12 | 3.3 (1.7–6.6) | 0.001 | 0.8 (0.3–2.6) | 0.765 |
| Type of LSCS | Elective | Emergency | 2.1 (1.3–3.4) | 0.003 | 3.0 (1.1–8.8) | 0.041 |
| No. of persons in OT | <5 | 5–10 | 0.3 (0.1–0.7) | 0.005 | 0.4 (0.1–1.1) | 0.078 |
| | | 11–15 | 0.3 (0.1–0.6) | 0.002 | 0.4 (0.1–1.3) | 0.145 |
| | | >15 | 0.2 (0.02–1.4) | 0.100 | 0.4 (0.04–4.1) | 0.456 |

## Discussion

In this large prospective study, the incidence of SSI was 4.6% with a single dose prophylactic antibiotic as per the WHO guidelines, detected within 30 days post-operative period as per CDC criteria. A single dose of prophylactic antibiotic, injection cefazolin was given and if not available, injection cefotaxime was given 30 minutes to one hour before skin incision for both elective and emergency caesarean sections. The incidence of SSI is low compared to most Indian studies, which have reported a burden ranging from 3.12% to as high as 24.2% over the last decade [7, 14–16, 19–21]. Most of these studies did not follow the WHO guidelines on pre-operative antibiotic use, and some even used multiple antibiotics in multiple doses before or after surgery [15, 19–21]. Therefore, we recommend adherence to the guidelines unless there is evidence-based justification to deviate from them.

We found that obesity is a strong risk factor for developing SSI. A meta-analysis has revealed that obese pregnant women have a higher risk of wound infections in all settings [22]. To prevent SSI in obese women, various guidelines recommend higher dose of antibiotics based on several studies [23–25]. However, these guidelines were advocated after commence-ment of our study. Based on these facts, we have changed our institutional policy to include double dose of antibiotic prophylaxis for obese women. We expect that this practice will fur-ther reduce the SSI burden in our setting.

Our study found that the younger (aged <25 years) are at high risk of developing SSI after CS. In India, inconsistent results are found in the literature in this regard [10, 14, 20, 26]. We found higher risk with emergency CS with borderline significance; but most studies in litera-ture have confirmed a higher risk of SSI with emergency CS [16, 27, 28].

In the present study, *Staphylococcus aureus*, a gram-positive organism, was the commonest organism isolated from caesarean wound infections. The other common organisms that were isolated include Klebsiella and E.coli, both being gram-negative organisms. The finding is con-sistent with the findings from other Indian studies which has reported co infections with both gram negative and gram-positive organisms [14–16, 20].

The strengths of the present study were that it was done prospectively, single dose antibiotic was prescribed as per the WHO guidelines, and the women were followed for 30 days post-operation as per the CDC criteria. Another strength of this study is that the proforma was filled by study nurses only, to rule out reporting bias. However, the study was based on a single cen-tre, and thus, the results might not be generalizable to the other contexts.

**Table 4. Sensitivity and resistance pattern of the organisms.**

| | Methicilin sensitive Staphylococcus aureus (n = 7) | Klebsiella (n = 13) | E. Coli (n = 12) | Citobacter (n = 4) | Acinetobacter (n = 3) | Proteus mirabilis (n = 1) | Providencia (n = 1) |
|---|---|---|---|---|---|---|---|
| Sensitivity (Number of isolates) | | | | | | | |
| Amoxyclav | 2 | 3 | 1 | 3 | | | 1 |
| Cefepime | 1 | 9 | | 2 | 1 | | 1 |
| Ceftazidime and Clavulanic acid | | 13 | | | | | |
| Ceftriaxone | | 4 | | 1 | 2 | | |
| Ciprofloxacin | | 8 | 1 | 2 | 1 | 1 | 1 |
| Co-Trimoxazole | 3 | 8 | 3 | 4 | 2 | | |
| Doxycycline | 2 | 2 | 1 | 2 | 1 | | 1 |
| Erythromycin | 3 | | | | | | |
| Gentamycin | | 3 | 1 | 1 | | | |
| Imipenem | | 10 | 10 | 3 | | 1 | 1 |
| Linezolid | 6 | | | | | | |
| Levofloxacin | 5 | | | | | | |
| Methicillin | 7 | | | | | | |
| Nitrofurantoin | | 1 | 6 | | | | |
| Norfloxacin | | 3 | 2 | 1 | | | |
| Piperacillin/ Tazobactum | 1 | 13 | 11 | 4 | 3 | 1 | 1 |
| Vancomycin | 7 | | | | | | |
| Resistance (Number of isolates) | | | | | | | |
| Amoxycillin | | 2 | 1 | 3 | 1 | | |
| Amoxyclav | 2 | 10 | 2 | 3 | 2 | 1 | |
| Amikacin | | 5 | 3 | 2 | 1 | 1 | |
| Ampicillin | 2 | 11 | 2 | 2 | | 1 | 1 |
| Aztreonam | 1 | 3 | 3 | 4 | | | |
| Cefepime | | 4 | 3 | 3 | | 1 | |
| Cefotaxime | | 5 | | 2 | | 1 | |
| Ceftriaxone | | | 3 | 3 | | | |
| Ciprofloxacin | 1 | 2 | 2 | | | | |
| Co-Trimoxazole | 2 | 3 | 1 | 1 | | 1 | |
| Nitrofurantoin | | 2 | | 1 | | | |
| Penicillin | 6 | | | | | | |

## Conclusion

The rate of SSI following caesarean deliveries subjected to single dose antibiotic prophylaxis was 4.6% with 99% being superficial infections. Young women and obese women were at high risk of developing SSI. Our finding also indicates the need for continuous vigilance on SSI control measures at the hospital-level.

## Supporting information

**S1 Questionnaire.**
(PDF)

**S1 Data.**
(XLSX)

## Acknowledgments

We are very grateful to our Study Nurses Ms Harati, Mrs Rosamma, study Co-ordinators Mrs. Deepa and Mrs Mamta and our IT team Mr. Purushotham Reddy and Mr. Narender.

## Author Contributions

**Conceptualization:** Neelima Angaali.

**Formal analysis:** Varun Agiwal, Nirupama A. Y.

**Methodology:** Kalpana Basany, Neelima Angaali.

**Project administration:** Kalpana Basany, Lakshmi Shailaja P.

**Supervision:** Kalpana Basany, Catherine Haggerty, P. S. Reddy.

**Validation:** Nirupama A. Y.

**Writing – original draft:** Kalpana Basany.

**Writing – review & editing:** Sirshendu Chaudhuri, Varun Agiwal, Shailendra D., P. S. Reddy.

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
