## [Decision Letter · Decision Letter 0]

12 Jun 2023

PONE-D-23-11380Dear Editor: -

Prospective cohort study of Surgical Site Infections following single dose antibiotic prophylaxis in caesarean section at a tertiary care teaching hospital in Medchal, IndiaPLOS ONE

Dear Dr. Kalpana,

Thank you for submitting your manuscript to PLOS ONE. After careful consideration, we feel that it has merit but does not fully meet PLOS ONE’s publication criteria as it currently stands. Therefore, we invite you to submit a revised version of the manuscript that addresses the points raised during the review process.

We look forward to receiving your revised manuscript.

Kind regards,

Arghya Das, MD

Academic Editor

PLOS ONE

Journal Requirements:

Additional Editor Comments:

Page 2, Line 34: “…………..profiles and the antimicrobial sensitivity and resistance pattern………..”

Comment: Replace ‘antimicrobial sensitivity and resistance pattern’ with ‘antimicrobial susceptibility’ only. Please use the word ‘susceptibility’ instead of ‘sensitivity’ throughout the manuscript.

Page 2, Line 49: “Of these, 92 participants……..”

Comment: Delete ‘Of these’ and start the sentence with ’Ninety-two participants’.

Page 2-3, Lines 54-55: “Microbial growth was observed in 75.8% 3 55 (n=50/66) samples”

Comment: Please rephrase the above sentence like “Microbial growth in culture was observed from 55 (75.8%) out of total 66 samples”. Please mention the statistical figures in and out of the parentheses within a sentence uniformly throughout the manuscript.

Page 3, Lines 55-56: The most common organisms identified were Staphylococcus aureus 56 (n=23, 46.0%), Klebsiella sp. (n=13, 26.0%), and Escherichia coli (n=12, 24.0%).

Comment: From the above sentence, it appears to be that the percentages of the organisms were calculated considering the total number of samples with a positive growth as denominator. Since multiple organisms have been isolated from 7 samples (as mentioned under Results), it will be helpful if the percentages are calculated considering total number of isolates in the denominator. Please also make relevant changes in the text under ‘Results’ section.

Page 3, Lines 58-61: Conclusion: Given the low rate of SSI following Caesarean deliveries subjected to single-dose antibiotic prophylaxis and the increased risk noted with obesity, it is rationale to practice the latest recommendations of WHO including higher dose for obese patients, unless there is compelling evidence to do otherwise in any context.

Comment: It is strongly advisable that the author should rewrite the conclusion of both Abstract and the main text based on the objective of the study only i.e. rate of CS and significant risk factors. Any recommendation based on the findings, may be included under the Discussion only.

Page 4, Lines 83-84: To prevent SSI, the WHO recommend using a single dose antibiotic, mostly the first generation cephalosporins, before 30 to minutes of incision for all women undergoing CS.

Comment: As per the reference cited, the recommendation for as single dose antibiotics is first generation cephalosporins or penicillin. Authors are advised to recheck the same and make required changes.

Page 5, Lines 107: We classified CS into two types- emergency and elective.

Comment: In general, emergency procedures have more risk for post-operative infections than elective procedures. As more than 60% of the patients underwent emergency CS it will be better if some comparative analysis is done based on the parameters described in the manuscript between emergency and elective CS.

Page 5, Line 113: For estimating 2% or less incidence of SSI after CS

Comment: Please clarify on the incidence of SSI after CS for calculating sample size. Or cite suitable reference.

Page 5, Lines 116-119: When Cefazolin was unavailable, injection Ampicillin 1 gram, OR injection Cefotaxim 1 gram was given.

Comment: Check the spelling of antibiotics. Few are written wrongly in the manuscript. Within a sentence, the name of the antibiotics should be written in lower case letters only.

As per the above-mentioned sentence, a third generation of cephalosporin was sometimes chosen if cefazolin, a first-generation cephalosporin, was not available. Is this also a part of hospital’s antibiotic policy?

Also mention under discussion why choosing a higher generation cephalosporins instead of first generation cephalosporins only will not cause any difference in rate of SSI in the study.

Page 6, Lines 121-124: Definition of variables: SSI was defined as- "An infection of the superficial or deep skin incision, or of an organ or space, occurring up to 30 days after surgery.” [17] There are two types of SSI- superficial and deep. As per the CDC definition, "A superficial SSI is infection identified during hospital stay or within 30 days……..

Comment: It is not clear what exact criteria were adopted for defining SSI. From the later sentences of the above paragraph, it seems to be that the authors have adopted CDC surveillance definitions. If NHSN-CDC surveillance definitions were considered to define the cases in the study, that should be directly cited as reference. Moreover, considering the detailed nature of the definitions of the guidelines it should be enough to state that “SSI for the present study was defined as per the CDC-NHSN guidelines”.

Page 7, Lines 150-151:  Data collection: All pertinent clinical details were collected by from the participants hospital record using structured questionnaire by a dedicated study nurse.

Comment: Please upload the structured questionnaire as supplementary file. Please also mention about the source of the questionnaire under the ‘Methods’ section.

Page 7, Lines 152-171: The hospital policy was to discharge the patient after suture removal………………….. microbiological procedure was followed for those who were discharged and followed up in the OPD.

Comment: Please provide some headings for the above paragraphs like “Laboratory diagnosis of SSI”, “Post-discharge Follow up”.

Page 10, Lines 200-202: The average hospital stay was 11.8 days (SD 3.7 days) in women who developed SSI and 9.3 (SD 3.1) days without SSI and mean difference was 2.5 days (95% CI: 1.9 to 3.2 days; p <0.001).

Comment: Please indicate under Statistical Methods in the Methods section about the specific test that was used for comparing the continuous quantitative variables.

Page 11, Lines 215-216: The only gram-positive organism, Staphylococcus, was commonly sensitive to- Linezolid (20 isolates, 87%)….

Comment: Authors need to recheck and confirm the susceptibility profile of staphylococcal isolates against linezolid. Linezolid resistance in Staphylococcus is still very low in India. No speciation has also been done for Staphylococcus. Please differentiate between Staphylococcus aureus and coagulase-negative staphylococcus in the manuscript.

Page 12, Lines 221-222: ……….. and Cefepime (9 isolates, 69.2%) and commonly resistant to Ampicillin (11 isolates, 222 84.6%) and Amoxycillin/Clavulanic acid combination (10 isolates, 76.9%).

Comment: It seems to be that the intrinsic resistance was not considered for some of the organisms in the study. Those with intrinsic resistance against some particular organism, need not to be analysed for resistance/susceptibility profile against that particular organisms. Authors are strongly encouraged to seek help from clinical microbiologists or ID specialists for these technical issues.

Page 16, Lines 246-247: Therefore, we recommend adherence to the guidelines unless there is evidence-based justification to deviate from them.

Comment: The statement may be overly correct, as the authors themselves did not uniformly followed the WHO guidelines (choice of antibiotics) for all the subjects.

Page 17, Lines 262-263: The finding is consistent with the findings from other Indian studies which has reported mixed infections with gram negative and gram-positive organisms. [14–16,20]

Comment: The authors should rephrase the above sentence as mixed infections term may be misleading and often refers to co-infections by both types of organisms mentioned above.

Page 17, Lines 266-267: Another strength of this study is that the proforma was filled by study nurses only, to rule out reporting bias.

Comment: Authors should also mention about the questionnaire filled up by the nurses under the ‘Methods’ section.

Table 1

Comment: Please define anaemia (as per Hb% etc) and hypertension (SBP, DBP, etc) within the table.

Table 2

Comment: Please mention the p values for individual variables with regards to the multivariable analysis. It’s also not clear whether authors choose to perform multivariable analysis for those variables which had a significant value in univariate analysis. Please clarify under the Statistical Methods

Tables 3 and 4

Comment: Both the tables may not be required. Either resistance or susceptibility profile may be presented. Moreover, the colour coding of the cells seem arbitrary without mentioning the level of susceptibility/resistance. Authors should remake the tables with mentioning of percentages instead of absolute numbers. Please seek help from a clinical microbiologist for depicting the susceptibility profile. Name of the organisms should be properly written following standard nomenclature.

Reviewers' comments:

Reviewer's Responses to Questions

**Comments to the Author**

1. Is the manuscript technically sound, and do the data support the conclusions?

Reviewer #1: Partly

Reviewer #2: Yes

2. Has the statistical analysis been performed appropriately and rigorously? 

Reviewer #1: No

Reviewer #2: Yes

3. Have the authors made all data underlying the findings in their manuscript fully available?

Reviewer #1: Yes

Reviewer #2: Yes

4. Is the manuscript presented in an intelligible fashion and written in standard English?

Reviewer #1: Yes

Reviewer #2: Yes

5. Review Comments to the Author

Reviewer #1: Dear Authors congratulation for the work that has been put in this paper , however few points needs more crisper reprenetaion like isolates and their antibiogram contain two table very big one , MRSA and MSSA have not been differantiated ,thir is no mention on inherant ressitant that might lead to the query for many who might not be aware of the things

Similiary in the risk factor table the refferance group needs to be defined clearly

and in discussion/result the Antibiotic that were used for surgical prophylaxis ,their timimg if it is discuused thotoghly would justify the title that has been used for the paper

Reviewer #2: The findings of the study have been presented in a rational and scientific manner using standard English. It is the opinion of the reviewer that sample size calculation formula, and its reference be added to the manuscript.

6. PLOS authors have the option to publish the peer review history of their article (what does this mean?). If published, this will include your full peer review and any attached files.

Reviewer #1: No

Reviewer #2: No

---

## [Author Response · Author response to Decision Letter 0]

7 Aug 2023

Response to Reviewers

Page 2, Line 34: “………….profiles and the antimicrobial sensitivity and resistance pattern………..”

Comment: Replace ‘antimicrobial sensitivity and resistance pattern’ with ‘antimicrobial susceptibility’ only. Please use the word ‘susceptibility’ instead of ‘sensitivity’ throughout the manuscript.

Response: Agree with the suggestion, replaced with the word ‘ susceptibility’.

Page 2, Line 49: “Of these, 92 participants……..”

Comment: Delete ‘Of these’ and start the sentence with ’Ninety-two participants’.

Response: Agree with the suggestion , changed.

Page 2-3, Lines 54-55: “Microbial growth was observed in 75.8% 3 55 (n=50/66) samples”

Comment: Please rephrase the above sentence like “Microbial growth in culture was observed from 55 (75.8%) out of total 66 samples”. Please mention the statistical figures in and out of the parentheses within a sentence uniformly throughout the manuscript.

Response: As suggested by the reviewers,the sentence was rephrased. The statistical figures were mentioned uniformly throughout the manuscript.

Page 3, Lines 55-56: The most common organisms identified were Staphylococcus aureus 56 (n=23, 46.0%), Klebsiella sp. (n=13, 26.0%), and Escherichia coli (n=12, 24.0%).

Comment: From the above sentence, it appears to be that the percentages of the organisms were calculated considering the total number of samples with a positive growth as denominator. Since multiple organisms have been isolated from 7 samples (as mentioned under Results), it will be helpful if the percentages are calculated considering total number of isolates in the denominator. Please also make relevant changes in the text under ‘Results’ section

Response: Thanks for the suggestion. We have done the calculation accordingly. Also, as suggested by the other reviewer, we have also reported the number of Staphylococcus aureus isolates and number of Coagulase negative Staphylococcus isolates. 

Page 3, Lines 58-61: Conclusion: Given the low rate of SSI following Caesarean deliveries subjected to single-dose antibiotic prophylaxis and the increased risk noted with obesity, it is rationale to practice the latest recommendations of WHO including higher dose for obese patients, unless there is compelling evidence to do otherwise in any context.

Comment: It is strongly advisable that the author should rewrite the conclusion of both Abstract and the main text based on the objective of the study only i.e. rate of CS and significant risk factors. Any recommendation based on the findings, may be included under the Discussion only.

Response: As suggested, conclusion was changed.

Page 4, Lines 83-84: To prevent SSI, the WHO recommend using a single dose antibiotic, mostly the first generation cephalosporins, before 30 to minutes of incision for all women undergoing CS.

Comment: As per the reference cited, the recommendation for as single dose antibiotics is first generation cephalosporins or penicillin. Authors are advised to recheck the same and make required changes

Response: Rechecked the reference no.8. Changes made

Page 5, Lines 107: We classified CS into two types- emergency and elective.

Comment: In general, emergency procedures have more risk for post-operative infections than elective procedures. As more than 60% of the patients underwent emergency CS it will be better if some comparative analysis is done based on the parameters described in the manuscript between emergency and elective CS

Response: As advised by the reviewers, comparative analysis was done between elective and emergency CS

Page 5, Line 113: For estimating 2% or less incidence of SSI after CS

Comment: Please clarify on the incidence of SSI after CS for calculating sample size. Or cite suitable reference.

Response: We thank the reviewer for the comment. According to the study conducted by Vijayan et al (2016), we determined that a minimum sample size of approximately 1500 women undergoing caesarean section would be necessary to estimate a 4.1% incidence of surgical site infections (SSI). This estimation took into account a significance level (α) of 0.05, a power (β) of 0.8, and a margin of error of 0.01.

Reference: Vijayan CP, Mohandas S, Nath AG, Surgical site infection following caesarean section in a teaching hospital. International Journal of Scientific Study. 2016;3(12);97-101.

Page 5, Lines 116-119: When Cefazolin was unavailable, injection Ampicillin 1 gram, OR injection Cefotaxim 1 gram was given.

Comment: Check the spelling of antibiotics. Few are written wrongly in the manuscript. Within a sentence, the name of the antibiotics should be written in lower case letters only.

As per the above-mentioned sentence, a third generation of cephalosporin was sometimes chosen if cefazolin, a first-generation cephalosporin, was not available. Is this also a part of hospital’s antibiotic policy?

Also mention under discussion why choosing a higher generation cephalosporins instead of first generation cephalosporins only will not cause any difference in rate of SSI in the study.

Response: Spelling of cefotaxime changed. Yes, it is the hospital policy to use third generation cephalosporin if first generation of cephalosporin was not available.

Page 6, Lines 121-124: Definition of variables: SSI was defined as- "An infection of the superficial or deep skin incision, or of an organ or space, occurring up to 30 days after surgery.” [17] There are two types of SSI- superficial and deep. As per the CDC definition, "A superficial SSI is infection identified during hospital stay or within 30 days……..

Comment: It is not clear what exact criteria were adopted for defining SSI. From the later sentences of the above paragraph, it seems to be that the authors have adopted CDC surveillance definitions. If NHSN-CDC surveillance definitions were considered to define the cases in the study, that should be directly cited as reference. Moreover, considering the detailed nature of the definitions of the guidelines it should be enough to state that “SSI for the present study was defined as per the CDC-NHSN guidelines”.

Response: As suggested , definition of SSI was changed accordingly, reference given

Page 7, Lines 150-151: Data collection: All pertinent clinical details were collected by from the participants hospital record using structured questionnaire by a dedicated study nurse.

Comment: Please upload the structured questionnaire as supplementary file. Please also mention about the source of the questionnaire under the ‘Methods’ section.

Response: The questionnaire was uploaded as supplementary file.

Page 7, Lines 152-171: The hospital policy was to discharge the patient after suture removal………………….. microbiological procedure was followed for those who were discharged and followed up in the OPD.

Comment: Please provide some headings for the above paragraphs like “Laboratory diagnosis of SSI”, “Post-discharge Follow up”.

Response: As advised by the reviewers, headings were provided.

Page 10, Lines 200-202: The average hospital stay was 11.8 days (SD 3.7 days) in women who developed SSI and 9.3 (SD 3.1) days without SSI and mean difference was 2.5 days (95% CI: 1.9 to 3.2 days; p <0.001).

Comment: Please indicate under Statistical Methods in the Methods section about the specific test that was used for comparing the continuous quantitative variables.

Response: We agree with the reviewers. As suggested, mentioned in the statistical methods

Page 11, Lines 215-216: The only gram-positive organism, Staphylococcus, was commonly sensitive to- Linezolid (20 isolates, 87%)….

Comment: Authors need to recheck and confirm the susceptibility profile of staphylococcal isolates against linezolid. Linezolid resistance in Staphylococcus is still very low in India. No speciation has also been done for Staphylococcus. Please differentiate between Staphylococcus aureus and coagulase-negative staphylococcus in the manuscript.

Response: We thank the author for the comment. We have discussed the same with the clinical Microbiologist and reanalyzed the figures. There were 7 isolates of Staphylococcus aureus and we have reported the sensitivity and resistance based on these 7 isolates. We have differentiated between the two organisms, viz. Staphylococcus aureus and coagulase-negative staphylococcus.

Page 12, Lines 221-222: ……….. and Cefepime (9 isolates, 69.2%) and commonly resistant to Ampicillin (11 isolates, 222 84.6%) and Amoxycillin/Clavulanic acid combination (10 isolates, 76.9%).

Comment: It seems to be that the intrinsic resistance was not considered for some of the organisms in the study. Those with intrinsic resistance against some particular organism, need not to be analysed for resistance/susceptibility profile against that particular organisms. Authors are strongly encouraged to seek help from clinical microbiologists or ID specialists for these technical issues

Response: We have discussed with the Clinical Microbiologist and considered the intrinsic resistance and removed those antibiotics. We have merged table 3 and table 4 as suggested by the other reviewers also. 

Page 16, Lines 246-247: Therefore, we recommend adherence to the guidelines unless there is evidence-based justification to deviate from them.

Comment: The statement may be overly correct, as the authors themselves did not uniformly followed the WHO guidelines (choice of antibiotics) for all the subjects.

Response: Yes. We understand that and we have reported that practicality to avoid any biased statement. 

Page 17, Lines 262-263: The finding is consistent with the findings from other Indian studies which has reported mixed infections with gram negative and gram-positive organisms. [14–16,20]

Comment: The authors should rephrase the above sentence as mixed infections term may be misleading and often refers to co-infections by both types of organisms mentioned above

Response: The sentence has been rephrased.

Page 17, Lines 266-267: Another strength of this study is that the proforma was filled by study nurses only, to rule out reporting bias.

Comment: Authors should also mention about the questionnaire filled up by the nurses under the ‘Methods’ section

Response: As suggested by the reviewers , mentioned under the ‘Methods’ section

---

## [Decision Letter · Decision Letter 1]

11 Sep 2023

PONE-D-23-11380R1

Prospective cohort study of Surgical Site Infections following single dose antibiotic prophylaxis in caesarean section at a tertiary care teaching hospital in Medchal, IndiaPLOS ONE

Dear Dr. Basany,

Thank you for submitting your manuscript to PLOS ONE. After careful consideration, we feel that it has merit but does not fully meet PLOS ONE’s publication criteria as it currently stands. Therefore, we invite you to submit a revised version of the manuscript that addresses the points raised during the review process.

We look forward to receiving your revised manuscript.

Kind regards,

Arghya Das, MD

Academic Editor

PLOS ONE

Journal Requirements:

Additional Editor Comments:

The exponentiated coefficients for logistic regression model are interpreted as odds ratios whereas for log-binomial regression they are relative risk ratios. The authors have mentioned that they have conducted logistic regression but mentioned relative risks instead of odds ratio. This needs clarification.

Please edit the title of the manuscript while resubmission and delete "Dear Editor".

Reviewers' comments:

Reviewer's Responses to Questions

**Comments to the Author**

1. If the authors have adequately addressed your comments raised in a previous round of review and you feel that this manuscript is now acceptable for publication, you may indicate that here to bypass the “Comments to the Author” section, enter your conflict of interest statement in the “Confidential to Editor” section, and submit your "Accept" recommendation.

Reviewer #3: (No Response)

Reviewer #4: (No Response)

2. Is the manuscript technically sound, and do the data support the conclusions?

Reviewer #3: No

Reviewer #4: Yes

3. Has the statistical analysis been performed appropriately and rigorously? 

Reviewer #3: No

Reviewer #4: I Don't Know

4. Have the authors made all data underlying the findings in their manuscript fully available?

Reviewer #3: Yes

Reviewer #4: Yes

5. Is the manuscript presented in an intelligible fashion and written in standard English?

Reviewer #3: No

Reviewer #4: Yes

6. Review Comments to the Author

Reviewer #3: The statistical analysis is not in accordance with the research question. The study variables are not well defined. The research question is whether there is a reduction in the SSI following single antibiotic prophylaxis but the analysis is on various risk factors rather.

The study population is not homogenous which includes both elective and emergency caesarean section group.

The elective section group included by the author also has patients in prolonged labour which is incorrect.

There is no comparator group. The institute's previous SSI atleast should be mentioned. The manuscript needs to be rewritten addressing all these inputs.

Reviewer #4: Page 12, Lines 219-220: ……….. and Cefepime (9 isolates, 69.2%) and commonly resistant to Ampicillin (11 isolates, 84.6%) and Amoxycillin/Clavulanic acid combination (10 isolates, 76.9%).

Comment: It seems that these lines depicting resistance/susceptibility profile of Ampicillin in Klebsiella species has not been modified as suggested by other reviewers. Authors are suggested to check the intrinsic resistance profile of Klebsiella species and accordingly make changes in the manuscript.

7. PLOS authors have the option to publish the peer review history of their article (what does this mean?). If published, this will include your full peer review and any attached files.

Reviewer #3: **Yes: **J. Yavana Suriya

Reviewer #4: **Yes: **ANWITA MISHRA M.D

---

## [Author Response · Author response to Decision Letter 1]

19 Oct 2023

Responses to the reviewer’s comments

Editors comment:

1. The exponentiated coefficients for logistic regression model are interpreted as odds ratios whereas for log-binomial regression they are relative risk ratios. The authors have mentioned that they have conducted logistic regression but mentioned relative risks instead of odds ratio. This needs clarification.

Response: We thank the editor for pointing it out. We have revised the terminology used from ‘logistic regression’ to ‘binomial regression”. Vide line number 174-176.

Editors comment: 

2. Please edit the title of the manuscript while resubmission and delete "Dear Editor".

Response: Thanks. We will ensure that while resubmitting the manuscript.

Reviewers # 3 comments

3. The statistical analysis is not in accordance with the research question. The research question is whether there is a reduction in the SSI following single antibiotic prophylaxis but the analysis is on various risk factors rather.

Response: We thank the reviewer for the comment. Please note that our primary objective was- to determine the incidence of post CS SSI following the adoption of single-dose antibiotic prophylaxis as recommended by WHO [8] in a tertiary care teaching hospital in Medchal, Telangana state, India. (Vide- line number 93-95).

Besides, we had two more secondary objectives- “Additionally, we looked at the risk factors of SSI and reported the bacteriological profiles and the antimicrobial susceptibility pattern of the culture positive isolates.” (Vide line number 99-101)

In our results, we have articulated the results in the same line- the incidence of SSI, the risk factors and the susceptibility pattern of the culture positive isolates. 

All the authors have gone through the results and we strongly feel that the statistical analysis is in accordance with the research objectives.

4. The study variables are not well defined.

Response: We have added the definitions of all other variables those were missed out in the previous draft with appropriate references. (Vide line number 139 to 145). Please note that we have revised the sequence of references as well while incorporating the additional references.

5. The study population is not homogenous which includes both elective and emergency caesarean section group.

Response: We thank the author for the comment. Please note that in a hospital setting, we expect both elective and emergency caesarean section group. Hence, we believe that the population is homogeneous. In addition, we strongly feel that type of surgery may be directly related to SSI. Hence, we have reported the primary outcome of interest (SSI incidence) separately for both these groups to ensure clarity in the results.

6. The elective section group included by the author also has patients in prolonged labour which is incorrect.

Response: We agree with the reviewer and we have done the necessary correction in table 3.

7. There is no comparator group. The institute's previous SSI at least should be mentioned.

Response: As our primary outcome was to calculate the SSI incidence, we did not require a comparator group. Also, taking a historical control has it’s own challenges, especially the robustness of the information. Hence, we feel that taking a comparator group might not help improving the interpretation of the present finding. However, to calculate the risk, we have used the comparator group.

Reviewers #4 comment

8. It seems that these lines depicting resistance/susceptibility profile of Ampicillin in Klebsiella species has not been modified as suggested by other reviewers. Authors are suggested to check the intrinsic resistance profile of Klebsiella species and accordingly make changes in the manuscript.

Response: We thank the reviewer for pointing it out. Comment of the earlier reviewer was “It seems to be that the intrinsic resistance was not considered for some of the organisms in the study. Those with intrinsic resistance against some particular organism, need not to be analysed for resistance/susceptibility profile against that particular organisms. Authors are strongly encouraged to seek help from clinical microbiologists or ID specialists for these technical issues.” 

Though we changed mostly based on the comment, we might have missed a few inadvertently. We have now incorporated the changes as suggested after discussing with the clinical microbiologist. The changes are as follows: 

• The ampicillin resistance pattern for Klebsiella has been removed from the table and from the text. (Vide table 4, line number 227-8 in the text)

• For Staphylococcus aureus (Table 4), we have removed Methicillin as the Methicillin susceptibility/resistance was screened by cefoxitin). We have also removed Aztreonam and Vancomycin resistance/susceptibility pattern for Staphylococcus aureus. (Vide table 4, line number 223).

---

## [Decision Letter · Decision Letter 2]

2 Nov 2023

PONE-D-23-11380R2Prospective cohort study of Surgical Site Infections following single dose antibiotic prophylaxis in caesarean section at a tertiary care teaching hospital in Medchal, IndiaPLOS ONE

Dear Dr. Basany,

Thank you for submitting your manuscript to PLOS ONE. After careful consideration, we feel that it has merit but does not fully meet PLOS ONE’s publication criteria as it currently stands. Therefore, we invite you to submit a revised version of the manuscript that addresses the points raised during the review process.

We look forward to receiving your revised manuscript.

Kind regards,

Arghya Das, MD

Academic Editor

PLOS ONE

Journal Requirements:

**Additional Editor Comments:**

The manuscript has been satisfactorily revised. Addressing few minor issues would make the manuscript flawless.

Reviewers' comments:

Reviewer's Responses to Questions

**Comments to the Author**

1. If the authors have adequately addressed your comments raised in a previous round of review and you feel that this manuscript is now acceptable for publication, you may indicate that here to bypass the “Comments to the Author” section, enter your conflict of interest statement in the “Confidential to Editor” section, and submit your "Accept" recommendation.

Reviewer #3: All comments have been addressed

Reviewer #4: All comments have been addressed

2. Is the manuscript technically sound, and do the data support the conclusions?

Reviewer #3: Yes

Reviewer #4: Yes

3. Has the statistical analysis been performed appropriately and rigorously? 

Reviewer #3: Yes

Reviewer #4: I Don't Know

4. Have the authors made all data underlying the findings in their manuscript fully available?

Reviewer #3: No

Reviewer #4: Yes

5. Is the manuscript presented in an intelligible fashion and written in standard English?

Reviewer #3: Yes

Reviewer #4: Yes

6. Review Comments to the Author

Reviewer #3: 1) Labouring patients are included in the elective section group in the table.

2) Grammatical and punctuation errors needs to be corrected.

3) The names of the antibiotics like methicillin, ceftazidime, etc. and terms like caesarean delivery, elective need not be capitalised.

Reviewer #4: Authors have mentioned that all the isolates of Staphylococcus were sensitive to Methicillin (Table 4). (line number 223), but have not shown the same in table 4. As per CLSI guidelines, though Methicillin susceptibility/resistance was screened by cefoxitin, it should be mentioned in the table and the isolate should be reported in results as Methicillin sensitive Staphylococcus aureus (MSSA).

7. PLOS authors have the option to publish the peer review history of their article (what does this mean?). If published, this will include your full peer review and any attached files.

Reviewer #3: **Yes: **Yavana Suriya.J

Reviewer #4: **Yes: **ANWITA MISHRA M.D

---

## [Author Response · Author response to Decision Letter 2]

12 Dec 2023

Responses to reviewers

Reviewers # 3 comments

1. Labouring patients are included in the elective section group in the table

Response: We thank the reviewer for pointing it out. We have gone through the data source carefully and learned that an inadvertent error took place from our end , and we misclassified a few participants as elective section group, whereas, they should have been included in the emergency CS group. Hence, we have updated the data and reanalyzed it and updated the results and the discussion section as per the need. We have also replaced the previous data that we submitted to PLOS One with the updated data.

2. Grammatical and punctuation errors needs to be corrected

 Response: Thank you the reviewer for pointing out. We have addressed them.

3. The names of the antibiotics like methicillin, ceftazidime, etc. and terms like caesarean delivery, elective need not be capitalised

 Response: As suggested, we have changed into small letters.

.

Reviewers #4 comment

Authors have mentioned that all the isolates of Staphylococcus were sensitive to Methicillin (Table 4). (line number 223), but have not shown the same in table 4. As per CLSI guidelines, though Methicillin susceptibility/resistance was screened by cefoxitin, it should be mentioned in the table and the isolate should be reported in results as Methicillin sensitive Staphylococcus aureus (MSSA)

Response: We thank the reviewer for the comment. We have made necessary changes in the table.

---

## [Editor Report · Decision Letter 3]

18 Dec 2023

PONE-D-23-11380R3Prospective cohort study of Surgical Site Infections following single dose antibiotic prophylaxis in caesarean section at a tertiary care teaching hospital in Medchal, IndiaPLOS ONE

Dear Dr. Basany,

Thank you for submitting your manuscript to PLOS ONE. After careful consideration, we feel that it has merit but does not fully meet PLOS ONE’s publication criteria as it currently stands. Therefore, we invite you to submit a revised version of the manuscript that addresses the points raised during the review process.

We look forward to receiving your revised manuscript.

Kind regards,

Arghya Das, MD

Academic Editor

PLOS ONE

Journal Requirements:

Additional Editor Comments:

Authors have made few changes in the statistical figures but similar changes have not been made in the Abstract.

As per Table 1, the total number of patients for whom data on labour was available are 2015. But the same has been mentioned as 2010 in Table 3.

In Table 3, the percentage of participants in the Emergency CS group with <6 hours duration of labour has been wrongly mentioned.

In Table 3, the number of participants in the Elective CS group (non-labour) should be 783 not 78.3

---

## [Author Response · Author response to Decision Letter 3]

19 Dec 2023

Responses to Editor comments

1. Authors have made few changes in the statistical figures but similar changes have not been made in the Abstract

Response: Thank the Editor for pointing out. Statistical figures have been changed.

2. As per Table 1, the total number of patients for whom data on labour was available are 2015. But the same has been mentioned as 2010 in Table 3.

Response: As suggested, the changes have been made.

3. In Table 3, the percentage of participants in the Emergency CS group with <6 hours duration of labour has been wrongly mentioned

Response: Thank the Editor for pointing out. We corrected the percentage.

4. In Table 3, the number of participants in the Elective CS group (non-labour) should be 783 not 78.3

Response: In Table 3, the number of participants in the Elective CS group was mentioned as 783.

---

## [Editor Report · Decision Letter 4]

22 Dec 2023

Prospective cohort study of Surgical Site Infections following single dose antibiotic prophylaxis in caesarean section at a tertiary care teaching hospital in Medchal, India

PONE-D-23-11380R4

Dear Dr. Basany,

We’re pleased to inform you that your manuscript has been judged scientifically suitable for publication and will be formally accepted for publication once it meets all outstanding technical requirements.

Kind regards,

Arghya Das, MD

Academic Editor

PLOS ONE

Additional Editor Comments (optional):

A small correction for the number of Staphylococcus isolates and its percentage needs to be made in the Abstract. This discrepancy is present between what is mentioned in the main text (number of S. aureus n=7) and in the Abstract (number of S. aureus isolates n=23). This correction has not been made since revision of the earlier version of manuscripts. Authors should change it during the authors proofing stage.
---

## [Editor Report · Acceptance letter]

17 Jan 2024

PONE-D-23-11380R4 

PLOS ONE

Dear Dr. Basany, 

I'm pleased to inform you that your manuscript has been deemed suitable for publication in PLOS ONE. Congratulations! Your manuscript is now being handed over to our production team.

Kind regards, 

on behalf of

Dr. Arghya Das 

Academic Editor

PLOS ONE